# Association of OCT biomarkers and visual impairment in patients with diabetic macular oedema with vitreomacular adhesion

**Brughanya Subramanian, Chitralekha Devishamani⬚, Rajiv Raman, Dhanashree Ratra⬚ ***

Department of Ophthalmology, Shri Bhagwan Mahavir Vitreoretinal Services, Sankara Nethralaya, Chennai, Tamil Nadu, India

\* dhanashreeratra@gmail.com

**Data Availability Statement:** Data is provided as supporting information file.

**Funding:** The authors received no specific funding for this work.

## Abstract

### Background

To analyse the distribution of spectral domain optical coherence tomography (SD-OCT) biomarkers in different types of vitreomacular adhesion (VMA) associated visual impairment in diabetic macular oedema.

### Methods

A total of 317 eyes of 202 patients were enrolled. Cases were divided into two groups focal VMA and broad VMA and subjects with no VMA were enrolled as controls. A grading platform was used for evaluating the morphology of diabetic macular oedema (DME), using good-quality SD-OCT images. Grading was done for VMA and the biomarkers. Best corrected visual acuity (BCVA), central retinal thickness (CRT) and central subfield thickness (CSFT) was also recorded.

### Results

The CRT (p = <0.001) and CSFT (p = <0.001) values were statistically significant between the groups. Except for Inner Nuclear Layer Cysts (p = <0.001), absence of Bridging Tissue that is composed of muller cell fibers and bipolar cells (p<0.001), and Hyper Reflective Dots (HRD) in cyst (p = 0.006) there were no significant differences in the distribution of OCT biomarkers among the 3 groups (focal VMA, broad VMA and no VMA). Only Disorganization of Retinal Inner Layers (DRIL) (p = 0.044) showed significant association with vision impairment in all the 3 groups.

### Conclusion

The distribution of OCT biomarkers was similar across all eyes of cases and controls. However, they were more likely to be associated with visual impairment in the presence of VMA than no VMA.

**Competing interests:** The authors have declared that no competing interests exist.

## Introduction

The vitreoretinal interface is an adhesive sheet that connects the posterior vitreous cortex to the internal limiting membrane [1]. With ageing, the adhesion between the posterior vitreous cortex and the inner limiting membrane becomes weak. It causes complete or incomplete posterior vitreous detachment (PVD), resulting in vitreomacular interface (VMI) diseases [2, 3]. VMI abnormalities are reported to occur in up to 7% to 16% of eyes with diabetic macular oedema (DME), with an annual incidence as high as 4.5% [4, 5]. VMI abnormalities are an umbrella term used to describe a series of disorders. These anomalies include vitreomacular traction (VMT), epiretinal membrane (ERM) and vitreomacular adhesion (VMA) [6–8].

VMA describes residual adhesion between the vitreous and macula, occurring within the context of an incomplete PVD. It might not lead to any retinal abnormality but may exert traction on the underlying macula, distorting the retinal architecture and leading to VMT [9]. Studies have shown variability in response to anti-VEGF therapy among eyes with DME associated with different subtypes of VMI abnormalities [4, 10, 11]. There is evidence that OCT-based biomarkers can act as an important factor to predict visual and anatomical outcomes in patients with DME and few of these OCT markers are associated with visual impairment in DME [12–15]. However, the distribution of these OCT markers in the presence of VMI abnormalities and their association with visual impairment has not been studied. Since VMA is one of the VRI abnormalities, this can also alter the retinal architecture [9]. It is necessary to look into the changes caused due to VMA for predicting better clinical outcomes.

This study aims to analyse the distribution of spectral domain optical coherence tomography (SD-OCT) biomarkers in different types of vitreomacular adhesion (VMA) and the associated visual impairment in diabetic macular oedema.

## Materials and methods

This was a case–control study held at a tertiary eye hospital in Chennai, South India. The Vision Research Foundation institutional review board approved the study on 5th April 2021, and data were accessed for research purposes from 10th April 2021. Since the study design is retrospective, the data were collected from the visit 1st January 2015 till 30th June 2022. All the research adhered to the tenets of the Helsinki Declaration. Written informed consent was obtained from the enrolled subjects, and all authors had access to subject information for data collection.

The study participants were categorised into two groups; cases in the presence of VMA (+) and controls in the absence of VMA (-). The presence of VMA in our study was identified according to The International Vitreomacular Traction Study Group classification [6]. Where the authors defined VMA according to following criteria 1) Evidence of perifoveal vitreous cortex detachment from retinal surface 2) Macular attachment of the vitreous cortex within a 3mm radius of the fovea 3) No detectable change in foveal contour or underlying retinal tissues. We followed the same International Vitreomacular Traction Study Group classification to further subdivide VMA (+) group in our study into two groups, either focal (<1500 mm) or broad (≥1500 mm), based on the size of the adhesion [6]. (Fig 1) The visual acuity of study participants was divided into no impairment in vision (BCVA between 6/6–6/9) and impaired vision (BCVA <6/12) for analysis according to Vision loss categories chosen by the Global Burden of Diseases (GBD) Vision Loss Expert Group [16].

### Inclusion and exclusion criteria

Subjects were included in the study if they met the following criteria: (1) Individuals (18 years or older) with type 2 diabetes mellitus and DME, (2) availability of SD-OCT scans of sufficient quality for grading, and (3) no other confounding ocular condition that could decrease visual

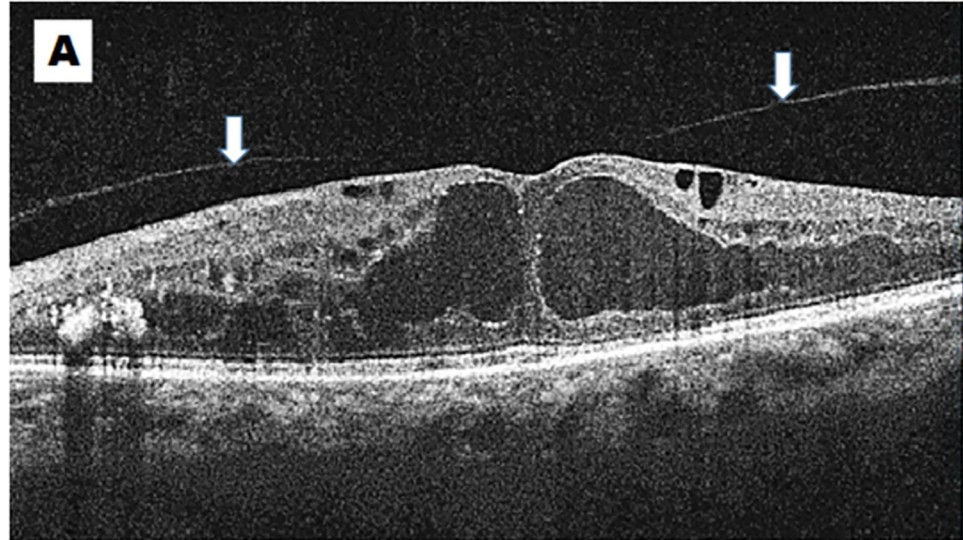

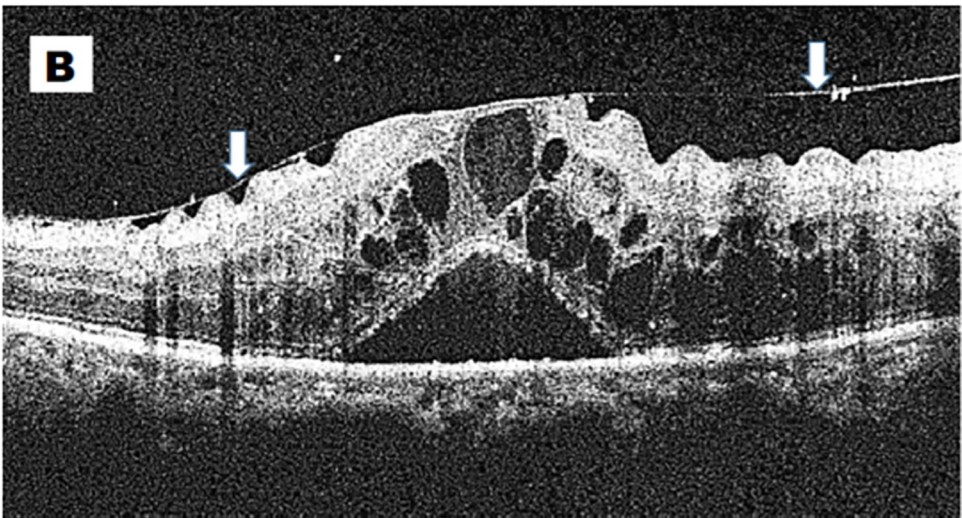

**Fig 1.** Optical coherence tomography of images of vitreomacular adhesion showing (A) focal adhesion (<1500mm) and (B) broad adhesion (≥1500mm). Arrows indicate vitreous adhesion to the central macula.

acuity other than DME. Subjects with the following criteria were excluded: (1) vitreomacular interface abnormalities besides VMA such as ERM, VMT and macular hole, (2) pre-existing retinal or macular disease other than DR or DME and (3) SD-OCT images with poor quality that was insufficient for assessment.

The medical records of all individuals were reviewed for baseline demographics, including age, gender, duration of DM, DR severity, cardiovascular co morbidities, dyslipidaemia, slit-lamp bio microscopy examination, intraocular pressure, the number and type of anti-VEGF injections given and dilated fundus evaluation. BCVA was measured with Snellen charts and converted to the Logarithm of the Minimum Angle of Resolution (Log MAR).

## OCT image analysis

SD-OCT (Cirrus HD-OCT, Carl, Zeiss Meditec Inc., Dublin, CA) was performed on each individual at baseline, and all OCT images were analysed by two graders (optometrist) and, in case of

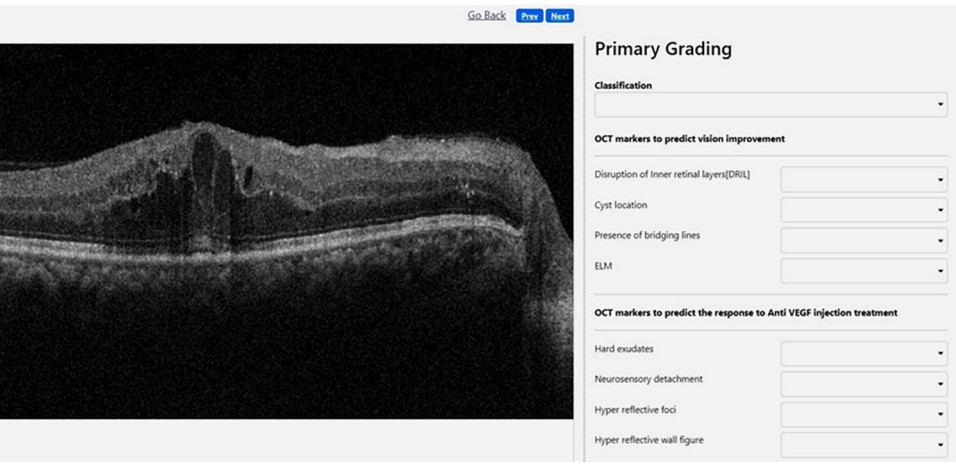

**Fig 2. Grading platform showing optical coherence tomography scan image with primary grading area.**

discrepancies examined by an expert (retina specialist). Quantitative measurement, the central retinal thickness (CRT) and central subfield thickness (CSFT) were automatically calculated by the built-in software version 6.0. Qualitative analysis of OCT images was done with a customised grading platform to grade the presence or absence of OCT biomarkers in DME (Fig 2).

## Definitions

**Grading of Diabetic retinopathy (DR).** Clinical grade of DR as noted by the treating ophthalmologist based on the International Clinical Diabetic Retinopathy (ICDR) disease severity scale was considered as the grade of DR [17].

**Type of VMA (focal/broad).** The size of adhesion classified the VMA into either focal (<1500 mm) or broad (≥1500 mm) [6].

## OCT markers to predict visual impairment

**The disorganisation of inner retinal layers (DRIL).** It was identified if there is a failure to recognise the boundaries of the ganglion cell layers-inner plexiform layer (GCL-IPL) complex, inner nuclear layer (INL), and the outer plexiform layer (OPL) within the central 1000 microns region of interest [18, 19].

**Intraretinal cyst.** It was defined as oval or circular lesions with reflectivity lower than or equal to the vitreous, found within the layers of Internal limiting membrane through external limiting membrane, with or without hyper reflective material within the lesion [12]. The cysts were assessed for their location, predominance in inner layers, outer layers or both.

**Presence of bridging retinal processes.** These are hyperreflective tissue between cystic cavities, believed to be composed of Muller cells fibers and bipolar cells, and are associated with good functional outcomes [12].

**Ellipsoid zone (EZ) and External limiting membrane (ELM) disruption.** The EZ marks the photoreceptor inner segment/outer segment (IS/OS) junction, and the external limiting membrane (ELM) is the junction between Muller cells and photoreceptor inner segments [12]. Discontinuity of EZ and ELM was noted as the presence of disruption.

## OCT markers to predict anatomical response to treatment

**Hard exudates.** Hard Exudates were identified as irregularly shaped, well-defined lesions of reflectivity equal to or greater than the RPE-Bruch's complex that cast a shadow on

structures beneath them of 30 microns or more in size, which corresponds to discrete yellow-white lipid deposits on fundus evaluation [12]. They were graded as predominance in central 1mm, outside central 1mm, or both.

**Hyper-reflective foci.**   These were identified as discrete round/ dot lesions with reflectivity similar to that of the retinal nerve fibre layer, of 30 microns or less in size, which is not associated with back shadowing or a corresponding detectable lesion [15]. The continuous or discontinuous ring of hyperreflective dots along the cyst (intraretinal cyst) inner wall was also noted.

**The neurosensory detachment.**   It was identified as separation the RPE- Bruch's complex and the interdigitation zone [12].

## Statistical analysis

Statistical analysis was done using statistical software SPSS, version 19.0 (IBM, Armonk, NY). Kruskal-Wallis test (for non-parametric data) was used to compare the control group, and VMA group variables and frequencies were compared using the Chi-square test (for categorical variables). A p-value less than 0.05 were considered statistically significant.

## Results

Table 1 presents the demographic and baseline characteristics of the participants in the study. The three groups, namely focal VMA, broad VMA, and the control group, exhibited similar distributions in terms of gender, duration of diabetes (focal VMA: 11years [8–20], broad VMA: 15 years [10–18] and control: 12 years [8–17]), presence of other systemic diseases, retinopathy grades, lens status, previous treatments, mean number of injections (focal VMA: 2 injections [1–3], broad VMA: 2 injections [1–3] and control: 2 injections [1–3]) and distance vision (focal VMA: 0.50 [0.20–0.70], broad VMA: 0.40 [0.20–0.60] and control: 0.30 [0.20–0.60]). However, there was a significant difference in observed in age (p = 0.020) among the three groups, focal VMA group had a higher median age of 63 years (57–67) years, compared to the broad VMA group 60 years (53–65), and control group 59 years (51–64). Additionally, the values of CRT (p<0.001) and CSFT (p<0.001) also displayed significant difference among the three groups (focal VMA, broad VMA and the control group).

Table 2 shows the distribution of OCT biomarkers between cases (focal VMA group and broad VMA group) and controls (no VMA group). The variables that were found to have a significant differences were cysts in the inner layer (p<0.01), absence of bridging tissue (p<0.01) and hyper reflective dots in the cyst wall (p = 0.006). Other OCT biomarkers such as DRIL (79.80%), cyst in the outer layer (10.57%), cyst in both layers (86.53%), disrupted ELM (68.26), disrupted EZ (68.26), hard exudates within 1mm (1.92), presence of NSD (31.73%), presence of HRF (96.15%) and presence of HR cyst wall (52.88%) were more frequently seen in the broad VMA group but it was not statistically significant.

Table 3 shows whether the presence of these biomarkers was associated with visual impairment in the study groups. Presence of DRIL was associated with visual impairment in all the three groups focal VMA (p = 0.004), broad VMA (p = <0.00) and no VMA (p = 0.044). In the VMA group, cyst location in both layers (focal VMA p = 0.006, broad VMA p = 0.006), discontinuous ELM (focal VMA p = 0.029, broad VMA p = <0.001), discontinuous EZ (focal VMA p = 0.029, broad VMA p = <0.001), presence of HE in both location (focal VMA p = 0.003, broad VMA p = 0.002), presence of HRF (focal VMA p = 0.003, broad VMA p = 0.001) and presence of HR cyst wall (focal VMA p = 0.014, broad VMA p = 0.043) were associated with vision impairment. Similarly, OCT biomarkers like cyst location in outer layer (p = 0.035), absence of bridging lines (p = <0.001) and presence of NSD (p = 0.024) were

**Table 1. Baseline characteristics of study groups.**

| Variables | VMA with Focal adhesion (n = 103) | VMA with Broad adhesion (n = 104) | No VMA (n = 110) | p |
|---|---|---|---|---|
| Median Age, years | 63 (57–67) | 60 (53–65) | 59 (51–64) | **0.020**[*†] |
| Gender, *n* (%) | | | | |
| Male | 69 (66.99) | 71 (68.26) | 84 (76.36) | 0.411[‡] |
| Female | 34 (33.00) | 33 (31.73) | 26 (23.63) | 0.542 |
| Median duration of DM, years | 11(8–20) | 15 (10–18) | 12 (8–17) | 0.310[†] |
| Hypertension, *n* (%) | 53 (51.45) | 54 (51.92) | 65 (59.09) | 0.462[‡] |
| Dyslipidemia, *n* (%) | 19 (18.44) | 20 (19.23) | 20 (18.18) | 0.983[‡] |
| Severity of DR, *n* (%) | | | | |
| Mild NPDR | 19 (18.44) | 7 (6.73) | 14 (12.72) | 0.066[‡] |
| Moderate NPDR | 26 (25.24) | 37 (35.57) | 26 (23.63) | 0.257 |
| Severe NPDR | 17 (16.50) | 18 (17.30) | 24 (21.81) | 0.482 |
| PDR | 41 (39.80) | 42 (40.38) | 46 (41.81) | 0.850 |
| Median IOP, mm Hg | 14 (12–16) | 14 (13–16) | 14 (12–16) | 0.135[†] |
| Lens status, *n* (%) | | | | |
| Phakia | 67 (65.04) | 78 (75) | 71(64.54) | 0.650[‡] |
| Pseudophakia | 35 (33.98) | 26 (25) | 39(35.45) | 0.264 |
| Treatment nature, *n* (%) | | | | |
| Treatment naive | 54 (52.42) | 59 (56.73) | 65 (59.09) | 0.600[‡] |
| Intravitreal therapy | 49 (47.57) | 45 (43.26) | 45 (40.90) | 0.891 |
| Median no of injections | 2 (1–3) | 2 (1–3) | 2 (1–3) | 0.968[†] |
| DVA, Median (IQR) | 0.50 (0.20–0.70) | 0.40 (0.20–0.60) | 0.30 (0.20–0.60) | 0.151[†] |
| NVA, Median (IQR) | 0.40 (0.30–0.70) | 0.30 (0.30–0.60) | 0.30 (0.30–0.50) | **0.030**[*†] |
| CRT, Median (IQR) | 410.00 (275.00–597.00) | 450.00 (308.00–571.00) | 337.50 (202.75–461.00) | **<0.001**[*†] |
| CSFT, Median (IQR) | 389.00 (301.00–550.00) | 407.00 (340.25–544.25) | 349.50 (265.50–446.25) | **<0.001**[*†] |

DM, diabetes mellitus; DR, diabetic retinopathy; NPDR, non-proliferative diabetic retinopathy; PDR, proliferative diabetic retinopathy; IOP, intra ocular pressure; PRP, pan retinal photocoagulation; DVA, distance visual acuity; NVA, near visual acuity; CRT, central retinal thickness

† Kruskal Wallis

‡ Chi Square

statistically significant and was exclusively noticed with vision impairment in the broad VMA group.

## Discussion

Clinical and pathological evidence suggests that the abnormalities in the vitreoretinal interface may play an essential role in the pathogenesis of DME [11]. DME may be worsened due to anatomical or physiological changes in the posterior vitreous [9]. Traction due to the residual cortical vitreous on the macula after PVD thickened posterior hyaloid or tractional proliferative membranes may worsen DME [20]. Likewise, the loculation of cytokines in the pre-macular vitreous may also exacerbate the DME [13]. On the other hand, abnormalities like VMA and PVD may present as clinically asymptomatic conditions which can only be detected by the OCT [21]. Recent studies have demonstrated that the presence of VMA and PVD may affect the visual and anatomic outcome in subjects receiving anti-VEGF injections for DME and AMD [11, 22–25]. However, the effect of different types of VMA on DME and its association with OCT biomarkers has not been reported previously.

The study results demonstrate that the distribution of OCT biomarkers is similar among all three groups. Still, the visual impairment was largely identified in subjects with a broad VMA

**Table 2. Distribution of biomarkers in the study groups.**

| Variables | VMA with Focal adhesion (n = 103) | VMA with Broad adhesion (n = 104) | No VMA (n = 110) | p ‡ |
|---|---|---|---|---|
| DRIL, n (%) | 80 (77.66) | 83 (79.80) | 71(64.54) | 0.607 |
| Cyst Outer layer, n (%) | 7 (6.79) | 11 (10.57) | 6 (5.45) | 0.417 |
| Cyst Inner layer, n (%) | 8 (7.76) | 3 (2.88) | 25(22.72) | <**0.001**[*] |
| Cyst in both layers, n (%) | 88 (85.43) | 90 (86.53) | 79(71.81) | 0.670 |
| Absence of bridging tissue, n (%) | 39 (37.86) | 80 (76.92) | 41 (37.27) | <**0.001**[*] |
| Discontinuous ELM, n (%) | 54 (52.42) | 71 (68.26) | 48 (43.63) | 0.085 |
| Discontinuous EZ, n (%) | 54 (52.42) | 71 (68.26) | 48 (43.63) | 0.085 |
| Hard exudates Within 1mm, n (%) | 0 (0) | 2 (1.92) | 1 (0.90) | 0.779 |
| Hard exudates More than 1mm, n (%) | 28 (27.18) | 35 (33.65) | 42 (38.18) | 0.247 |
| Hard exudates Both, n (%) | 74 (71.84) | 67 (64.42) | 67 (60.90) | 0.790 |
| Presence of NSD, n (%) | 32 (31.06) | 33 (31.73) | 28 (25.45) | 0.798 |
| Presence of HRF, n (%) | 100 (96.15) | 100 (96.15) | 107(97.27) | 0.852 |
| Presence of HR dots in the cyst wall, n (%) | 54 (51.92) | 55 (52.88) | 86 (78.18) | **0.006**[*] |

DRIL, disorganization of retinal inner layers; ELM, external limiting membrane; EZ, ellipsoid zone; CSFT, central subfield thickness; NSD, neurosensory detachment; HRF, hyperreflective foci; HR, hyper reflective ‡ Chi Square

group than the other groups. Vitreoretinal interface abnormalities (VRI) play an important role in pathophysiology of macular disease. We noticed that the CRT and CSFT values were higher among the broad VMA group when compared to the focal VMA and control groups. This is an interesting finding. VMA is generally thought to not contribute much to the retinal thickening, but the retinal thickening in VMA associated with DME is assumed to be caused due to the up regulation of VEGF and vascular hyper permeability [11].

Upon further analysis of OCT biomarkers, we found DRIL in most eyes with visual impairment, irrespective of the presence of VMA. Sun et al (2014), described DRIL as the

**Table 3. Visual impairment, OCT biomarkers and VMA characteristics in the study groups.**

| Variables | VMA with Focal adhesion (n = 103) | | | VMA with Broad adhesion (n = 104) | | | No VMA (n = 110) | | |
|---|---|---|---|---|---|---|---|---|---|
| | NI | Impaired | p ‡ | NI | Impaired | p ‡ | NI | Impaired | p ‡ |
| DRIL, n (%) | 27(26.21) | 53 (51.45) | **0.004**[*] | 27 (23.07) | 56 (53.84) | <**0.001**[*] | 27 (24.54) | 44 (40) | **0.044**[*] |
| Cyst Outer layer, n (%) | 3 (2.91) | 4 (3.88) | 0.705 | 2 (1.92) | 9 (8.65) | **0.035**[*] | 4 (3.63) | 2 (1.81) | 0.414 |
| Cyst Inner layer, n (%) | 2 (1.94) | 6 (5.82) | 0.157 | 1 (0.96) | 2 (1.92) | 0.564 | 13 (11.81) | 12 (10.90) | 0.841 |
| Cyst in both layers, n (%) | 31 (30.09) | 57 (55.33) | **0.006**[*] | 32 (30.76) | 58 (55.76) | **0.006**[*] | 35 (31.81) | 44 (40.00) | 0.311 |
| Absence of bridging tissue, n (%) | 18 (17.47) | 21 (20.38) | 0.631 | 25 (24.03) | 55 (52.88) | <**0.001**[*] | 25 (22.72) | 16 (14.54) | 0.160 |
| Discontinuous ELM, n (%) | 19 (18.44) | 35 (33.98) | **0.029**[*] | 21 (20.19) | 50 (48.07) | <**0.001**[*] | 19(17.27) | 29 (26.36) | 0.149 |
| Discontinuous EZ, n (%) | 19 (18.44) | 35 (33.98) | **0.029**[*] | 21 (20.19) | 50 (48.07) | <**0.001**[*] | 19 (17.27) | 29 (26.36) | 0.149 |
| Hard exudates Within 1mm, n (%) | 0 (0) | 0 (0) | - | 0 (0) | 2 (1.92) | - | 0(0) | 1 (0.90) | - |
| Hard exudates More than 1mm, n (%) | 11 (10.67) | 17 (16.50) | 0.257 | 12 (11.53) | 23 (22.11) | 0.063 | 24 (21.81) | 18 (16.36) | 0.355 |
| Hard exudates Both, n (%) | 24 (28.15) | 50 (43.68) | **0.003**[*] | 23 (22.11) | 44 (42.30) | **0.002**[*] | 28 (25.45) | 39 (35.45) | 0.179 |
| Presence of NSD, n (%) | 14 (13.59) | 18 (17.47) | 0.480 | 10 (9.61) | 23 (22.11) | **0.024**[*] | 10 (9.09) | 18(16.36) | 0.131 |
| Presence of HRF, n (%) | 35 (33.98) | 65 (63.10) | **0.003**[*] | 34 (32.69) | 66 (63.46) | **0.001**[*] | 50 (45.45) | 57 (51.81) | 0.499 |
| Presence of HR dots in the cyst wall, n (%) | 18 (17.47) | 36 (34.95) | **0.014**[*] | 20 (19.23) | 35 (33.65) | **0.043**[*] | 35 (31.81) | 51 (46.36) | 0.084 |

NI, Not Impaired; DRIL, disorganization of retinal inner layers; ELM, external limiting membrane; EZ, ellipsoid zone; CSFT, central subfield thickness; NSD, neurosensory detachment; HRF, hyper reflective foci; HR, hyper reflective
‡ Chi Square

surrogate structural biomarker to predict visual acuity in patients with existing or resolved Centre-involving DME [18]. DRIL can be considered an essential and independent biomarker for poor visual recovery [14]. The distribution of intraretinal cysts was found on the retina's outer and inner layer in all three groups; however, the broad VMA group showed most eyes affected with vision impairment with intraretinal cysts. In a study conducted by Al Fran et al (2014), it was found that the presence of bridging tissue between cystic cavities is linked to improved functional outcomes after receiving bevacizumab injections. The absence of the bridging process, on the other hand, is associated with poorer outcomes, and leading to retinal thinning and atrophy [26]. In our current study, we observed higher percentage of eyes (76.92%) within the broad VMA group that lacked bridging lines. These findings align with the previous literature and suggest that the outcomes for the broad VMA group are likely to be unfavourable, as indicated by the absence of bridging tissue.

Visual acuity is positively correlated with the survival rate of ELM and with the EZ [18, 27]. We found 68.26% from the broad VMA group, 52.42% from the focal VMA group and 43.63% from the control group with disrupted ELM and EZ. But the vision impairment was severe (p = <0.001) in the broad VMA group. Likewise, a few studies [27–32] state that the location of hard exudates at the macular region and the presence of NSD, HRF and HR cyst wall affect the vision severely, and we found the distribution of these biomarkers to be similar in all three groups. However, vision impairment was associated only with the VMA group, especially the broad VMA group.

VMA can be associated with various known retinal diseases that cause visual dysfunction [6, 33–40]. Several studies provide evidence supporting this hypothesis, suggesting that the occurrence of specific retinal complications is significantly higher in eyes with VMA compared to those with vitreomacular separation [6, 33–40]. This finding aligns with the widely acceptable understanding that VMA eyes exhibit stronger and persistent adhesions at fovea, resulting in detrimental complications [6, 33–40] Our study findings were also consistent with the previous literature. We observed a similar prevalence of biomarkers in eyes with or without VMA, but the presence of these biomarkers was more likely to be associated with visual impairment in eyes with VMA. Additionally, in the presence of VMA, the biomarkers appeared to have a greater extent and severity, which predisposed the eyes to a poorer visual prognosis. Hence, VMA can be considered an additional biomarker indicative of a poor visual prognosis.

Therefore, it is crucial to emphasize accurate diagnosis, long-term monitoring, and appropriate management when dealing with patients who experience adverse consequences associated with vitreomacular adhesion (VMA). In this regard, spectral domain optical coherence tomography (SD-OCT) emerges as a clinically valuable and readily available method that can identify specific abnormalities at the vitreomacular interface that might not be detectable using ultrasonography or indirect ophthalmoscopy [6, 33, 37, 41]. The study's strength is that we investigated multiple OCT-based biomarkers in a large group of DME patients with and without VMA. There are also some limitations in the current study. Firstly, the study nature is retrospective. Secondly, the study evaluated only the presence and absence of these biomarkers and not their severity, which might influence baseline visual acuity.

## Recommendations

As mentioned earlier, VMA may contribute to the development of various retinal complications. To ensure early detection and effective management of subsequent visual complications, it is advisable to closely monitor VMA in the long term. Future studies can be conducted prospectively to analyse the progression of VMA, particularly in terms of identifying any traction

leading to macular holes. When general ophthalmologists encounter patients with VMA, it is recommended to schedule regular follow-ups, especially utilizing tests like monthly SD-OCT, if available, to specifically evaluate the vitreomacular interface. If any significant visual dysfunction or retinal complications are identified, it is important to promptly refer the patient to a vitreoretinal specialist.

## Conclusion

The presence of VMA and its extent influence the presence and distribution of OCT- biomarkers in DME. The presence of these biomarkers in the presence of VMA has an influence on visual acuity in DME.

## Supporting information

**S1 Data.**
(XLSX)

## Author Contributions

**Conceptualization:** Rajiv Raman, Dhanashree Ratra.

**Data curation:** Brughanya Subramanian, Chitralekha Devishamani.

**Formal analysis:** Brughanya Subramanian.

**Investigation:** Brughanya Subramanian, Chitralekha Devishamani.

**Methodology:** Brughanya Subramanian, Chitralekha Devishamani, Rajiv Raman, Dhanashree Ratra.

**Project administration:** Rajiv Raman.

**Resources:** Brughanya Subramanian.

**Supervision:** Rajiv Raman, Dhanashree Ratra.

**Writing – original draft:** Brughanya Subramanian.

**Writing – review & editing:** Rajiv Raman, Dhanashree Ratra.

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
