## [Decision Letter · Decision Letter 0]

19 Jun 2023

PONE-D-23-11816Association of OCT biomarkers and visual impairment in patients with Diabetic Macular Oedema with Vitreomacular AdhesionPLOS ONE

Dear Dr. Ratra,

Thank you for submitting your manuscript to PLOS ONE. After careful consideration, we feel that it has merit but does not fully meet PLOS ONE’s publication criteria as it currently stands. Therefore, we invite you to submit a revised version of the manuscript that addresses the points raised during the review process.

We look forward to receiving your revised manuscript.

Kind regards,

Karim Adly Raafat

Academic Editor

PLOS ONE

Journal Requirements:

“none”

“none”

Additional Editor Comments:

Thank you for submitting this article 

Minor Revision needed

Abbreviations like HRD,  DRILL  , although mentioned and explained later in the test , they need to be clarified earlier in the abstract.

**Line 12  :** inner layers cysts need more clarification ( Inner Nuclear Layer INL , Inner plexiform Layers IPL or Ganglion Cell Layer GCC )

**Line 13 :** bridging tissue , needed earlier explanation , Muller cell fibers , Muller cell Pillars

**Line 98**:  DRIL definition needs revision Disorganization of Retinal Inner Layers , NOT Disruption  and this applies to the whole text . The original definition of DRIL also included disorganization of Retinal inner layers in central 1000 microns . The need revision in the test and also in the methodology

**Line 137 :** difference in the age …. Needs more clarification which group older or younger age

**Line 138 :** need adding units (years ) and explanation ( mean , range )

**Line 140 / 148 :** results need more clarification of the different groups

**Table 2  :** DRIL  Disorganization NOR disruption of Retinal Inner layers

**Line 198 :** VMA by definition NOT causing any traction . If there is antero posterior or tangential traction , it is then Vitreo macular Traction ( VMT) and not VMA. Inclusion criteria mentioned VMA and not VMT .

**Line 202 :** Sun et al missing year , this applied to all other references

**Line 204 :** Fovea involving DME  , the correct term Center – involving DME

In discussion  more explanation needed to explain how VMA WITHOUT VMT can affect vision

Reviewers' comments:

Reviewer's Responses to Questions

**Comments to the Author**

1. Is the manuscript technically sound, and do the data support the conclusions?

Reviewer #1: Yes

2. Has the statistical analysis been performed appropriately and rigorously? 

Reviewer #1: I Don't Know

3. Have the authors made all data underlying the findings in their manuscript fully available?

Reviewer #1: Yes

4. Is the manuscript presented in an intelligible fashion and written in standard English?

Reviewer #1: Yes

5. Review Comments to the Author

Reviewer #1: An interesting retrospective study with some new observations.

Explain the following abbreviations in the abstract : HR dots and DRIL.

Kindly review lines 210-212.

Add recommendations at the end.

6. PLOS authors have the option to publish the peer review history of their article (what does this mean?). If published, this will include your full peer review and any attached files.

Reviewer #1: No

---

## [Author Response · Author response to Decision Letter 0]

4 Jul 2023

Journal Requirements:

Response: Manuscript edited according to journal style

“none”

Response: Statement mentioned in the cover letter- as The authors received no specific funding for this work.” 

“none”

Response: Statement mentioned in the cover letter - The authors have declared that no competing interests exist.

Response: There are no restrictions in sharing the data and excel data supporting our study is provided as supporting information files as S1 Data (XLS)

Response: The name of the IRB is Vision Research Foundation and changes are mentioned at line 68

Response: References reviewed there are no retracted articles cited in this manuscript. The reference list has been modified to incorporate additional articles during the revision process. 

Additional Editor Comments:

Thank you for submitting this article 

Minor Revision needed

Abbreviations like HRD, DRILL, although mentioned and explained later in the test , they need to be clarified earlier in the abstract.

Response: Abbreviations are clarified 

Line 12 : inner layers cysts need more clarification ( Inner Nuclear Layer INL , Inner plexiform Layers IPL or Ganglion Cell Layer GCC )

Response: Clarification given 

Line 13 : bridging tissue , needed earlier explanation , Muller cell fibers , Muller cell Pillars

Response: Explanation given 

Line 98: DRIL definition needs revision Disorganization of Retinal Inner Layers , NOT Disruption and this applies to the whole text . The original definition of DRIL also included disorganization of Retinal inner layers in central 1000 microns . The need revision in the test and also in the methodology

Response: Revision done and changes are mentioned in methodology.

Line 137 : difference in the age …. Needs more clarification which group older or younger age

Response: Clarification given 

Response: Table 1 presents the demographic and baseline characteristics of the participants in the study. The three groups, namely focal VMA, broad VMA, and the control group, exhibited similar distributions in terms of gender, duration of diabetes (focal VMA: 11years [8-20], broad VMA: 15 years [10-18] and control: 12 years [8-17]), presence of other systemic diseases, retinopathy grades, lens status, previous treatments, mean number of injections (focal VMA: 2 injections [1-3], broad VMA: 2 injections [1-3] and control: 2 injections [1-3]) and distance vision (focal VMA: 0.50 [0.20-0.70], broad VMA: 0.40 [0.20-0.60] and control: 0.30 [0.20-0.60]). However, there was a significant difference in observed in age (p=0.020) among the three groups, focal VMA group had a higher median age of 63 years (57-67) years, compared to the broad VMA group 60 years (53-65), and control group 59 years (51-64). Additionally, the values of CRT (p<0.001) and CSFT (p<0.001) also displayed significant difference among the three groups (focal VMA, broad VMA and the control group).

Line 138: need adding units (years) and explanation (mean, range)

Response: Clarification given 

Response: Table 1 presents the demographic and baseline characteristics of the participants in the study. The three groups, namely focal VMA, broad VMA, and the control group, exhibited similar distributions in terms of gender, duration of diabetes (focal VMA: 11years [8-20], broad VMA: 15 years [10-18] and control: 12 years [8-17]), presence of other systemic diseases, retinopathy grades, lens status, previous treatments, mean number of injections (focal VMA: 2 injections [1-3], broad VMA: 2 injections [1-3] and control: 2 injections [1-3]) and distance vision (focal VMA: 0.50 [0.20-0.70], broad VMA: 0.40 [0.20-0.60] and control: 0.30 [0.20-0.60]). However, there was a significant difference in observed in age (p=0.020) among the three groups, focal VMA group had a higher median age of 63 years (57-67) years, compared to the broad VMA group 60 years (53-65), and control group 59 years (51-64). Additionally, the values of CRT (p<0.001) and CSFT (p<0.001) also displayed significant difference among the three groups (focal VMA, broad VMA and the control group).

Line 140 / 148: results need more clarification of the different groups

Response: Clarification given 

Response: Table 1 presents the demographic and baseline characteristics of the participants in the study. The three groups, namely focal VMA, broad VMA, and the control group, exhibited similar distributions in terms of gender, duration of diabetes (focal VMA: 11years [8-20], broad VMA: 15 years [10-18] and control: 12 years [8-17]), presence of other systemic diseases, retinopathy grades, lens status, previous treatments, mean number of injections (focal VMA: 2 injections [1-3], broad VMA: 2 injections [1-3] and control: 2 injections [1-3]) and distance vision (focal VMA: 0.50 [0.20-0.70], broad VMA: 0.40 [0.20-0.60] and control: 0.30 [0.20-0.60]). However, there was a significant difference in observed in age (p=0.020) among the three groups, focal VMA group had a higher median age of 63 years (57-67) years, compared to the broad VMA group 60 years (53-65), and control group 59 years (51-64). Additionally, the values of CRT (p<0.001) and CSFT (p<0.001) also displayed significant difference among the three groups (focal VMA, broad VMA and the control group).

Table 2 : DRIL Disorganization NOR disruption of Retinal Inner layers

Response: Changes are done wherever required 

Line 198 : VMA by definition NOT causing any traction . If there is antero posterior or tangential traction , it is then Vitreo macular Traction ( VMT) and not VMA. Inclusion criteria mentioned VMA and not VMT .

Response: Changes mentioned in line 243-248 

We noticed that the CRT and CSFT values were higher among the broad VMA group when compared to the focal VMA and control groups. This is an interesting finding. VMA is generally thought to not contribute much to the retinal thickening, but the retinal thickening in VMA associated with DME is assumed to be caused due to the up regulation of VEGF and vascular hyper permeability. [11] 

Line 202 : Sun et al missing year , this applied to all other references

Response: Changes are done in line 250

Line 204 : Fovea involving DME , the correct term Center – involving DME

Response: Changes are mentioned in line 252

In discussion more explanation needed to explain how VMA WITHOUT VMT can affect vision

Response: Clarification given in line 272-283

VMA can be associated with various known retinal diseases that cause visual dysfunction. [6,33-40] Several studies provide evidence supporting this hypothesis, suggesting that the occurrence of specific retinal complications is significantly higher in eyes with VMA compared to those with vitreomacular separation. [6,33-40] This finding aligns with the widely acceptable understanding that VMA eyes exhibit stronger and persistent adhesions at fovea, resulting in detrimental complications. [6,33-40] Our study findings were also consistent with the previous literature. We observed a similar prevalence of biomarkers in eyes with or without VMA, but the presence of these biomarkers was more likely to be associated with visual impairment in eyes with VMA. Additionally, in the presence of VMA, the biomarkers appeared to have a greater extent and severity, which predisposed the eyes to a poorer visual prognosis. Hence, VMA can be considered an additional biomarker indicative of a poor visual prognosis. 

Therefore, it is crucial to emphasize accurate diagnosis, long-term monitoring, and appropriate management when dealing with patients who experience adverse consequences associated with vitreomacular adhesion (VMA). In this regard, spectral domain optical coherence tomography (SD-OCT) emerges as a clinically valuable and readily available method that can identify specific abnormalities at the vitreomacular interface that might not be detectable using ultrasonography or indirect ophthalmoscopy. [6,33,37,41]

Comments to the Author

1. Is the manuscript technically sound, and do the data support the conclusions?

Reviewer #1: Yes

Response: Thank you for the response

2. Has the statistical analysis been performed appropriately and rigorously?

Reviewer #1: I Don't Know

3. Have the authors made all data underlying the findings in their manuscript fully available?

Reviewer #1: Yes

Response: Thank you for the response

4. Is the manuscript presented in an intelligible fashion and written in standard English?

Reviewer #1: Yes

Response: Thank you for the response

5. Review Comments to the Author

Reviewer #1: An interesting retrospective study with some new observations.

Explain the following abbreviations in the abstract : HR dots and DRIL.

Kindly review lines 210-212.

Add recommendations at the end.

Response: Thank you for the response, the explanation for HR Dots and DRILL is mentioned in abstract 

Revision for the lines 210-212 is as follows: 

Al Fran et al (2014), it was found that the presence of bridging tissue between cystic cavities is linked to improved functional outcomes after receiving bevacizumab injections. The absence of the bridging process, on the other hand, is associated with poorer outcomes, and leading to retinal thinning and atrophy. [26] In our current study, we observed higher percentage of eyes (76.92%) within the broad VMA group that lacked bridging lines. These findings align with the previous literature and suggest that the outcomes for the broad VMA group are likely to be unfavourable, as indicated by the absence of bridging tissue. 9These changes are done in line 256-263) 

Recommendations: 

As mentioned earlier, VMA may contribute to the development of various retinal complications. To ensure early detection and effective management of subsequent visual complications, it is advisable to closely monitor VMA in the long term. Future studies can be conducted prospectively to analyse the progression of VMA, particularly in terms of identifying any traction leading to macular holes. When general ophthalmologists encounter patients with VMA, it is recommended to schedule regular follow-ups, especially utilizing tests like monthly SD-OCT, if available, to specifically evaluate the vitreomacular interface. If any significant visual dysfunction or retinal complications are identified, it is important to promptly refer the patient to a vitreoretinal specialist.

6. PLOS authors have the option to publish the peer review history of their article (what does this mean?). If published, this will include your full peer review and any attached files.

Do you want your identity to be public for this peer review? For information about this choice, including consent withdrawal, please see our Privacy Policy.

Reviewer #1: No

---

## [Editor Report · Decision Letter 1]

6 Jul 2023

Association of OCT biomarkers and visual impairment in patients with Diabetic Macular Oedema with Vitreomacular Adhesion

PONE-D-23-11816R1

Dear Dr. Ratra,

We’re pleased to inform you that your manuscript has been judged scientifically suitable for publication and will be formally accepted for publication once it meets all outstanding technical requirements.

Kind regards,

Karim Adly Raafat, M.D.

Academic Editor

PLOS ONE
---

## [Editor Report · Acceptance letter]

10 Jul 2023

PONE-D-23-11816R1 

Association of OCT biomarkers and visual impairment in patients with Diabetic Macular Oedema with Vitreomacular Adhesion 

Dear Dr. Ratra:

I'm pleased to inform you that your manuscript has been deemed suitable for publication in PLOS ONE. Congratulations! Your manuscript is now with our production department. 

Kind regards, 

on behalf of

Professor Karim Adly Raafat 

Academic Editor

PLOS ONE